# A Spatial-Motion-Segmentation Algorithm by Fusing EDPA and Motion Compensation

**DOI:** 10.3390/s22186732

**Published:** 2022-09-06

**Authors:** Xinghua Liu, Yunan Zhao, Lei Yang, Shuzhi Sam Ge

**Affiliations:** 1School of Electrical Engineering, Xi’an University of Technology, Xi’an 710048, China; 2Department of Electrical and Computer Engineering, National University of Singapore, Singapore 119077, Singapore

**Keywords:** event camera, motion segmentation, motion compensation, depth estimation, motion flow

## Abstract

Motion segmentation is one of the fundamental steps for detection, tracking, and recognition, and it can separate moving objects from the background. In this paper, we propose a spatial-motion-segmentation algorithm by fusing the events-dimensionality-preprocessing algorithm (EDPA) and the volume of warped events (VWE). The EDPA consists of depth estimation, linear interpolation, and coordinate normalization to obtain an extra dimension (*Z*) of events. The VWE is conducted by accumulating the warped events (i.e., motion compensation), and the iterative-clustering algorithm is introduced to maximize the contrast (i.e., variance) in the VWE. We established our datasets by utilizing the event-camera simulator (ESIM), which can simulate high-frame-rate videos that are decomposed into frames to generate a large amount of reliable events data. Exterior and interior scenes were segmented in the first part of the experiments. We present the sparrow search algorithm-based gradient ascent (SSA-Gradient Ascent). The SSA-Gradient Ascent, gradient ascent, and particle swarm optimization (PSO) were evaluated in the second part. In Motion Flow 1, the SSA-Gradient Ascent was 0.402% higher than the basic variance value, and 52.941% faster than the basic convergence rate. In Motion Flow 2, the SSA-Gradient Ascent still performed better than the others. The experimental results validate the feasibility of the proposed algorithm.

## 1. Introduction

As a new type of sensing imaging device, event cameras, such as the dynamic vision sensor (DVS) or dynamic and active-pixel vision sensor (DAVIS) [1,2], are different from traditional cameras that shoot scenes or objects at a fixed frame rate [3]. The moving objects and light-intensity changes are focused on by event cameras. In general, the intensity change of each pixel is transmitted by a stream of asynchronous events. The information carried by an event includes the pixel coordinates (x,y), trigger time (t), and positive- or negative-event polarity (p). Compared with traditional cameras, event cameras have many advantages, for example, latency in microseconds, low motion blur, and high dynamic range (HDR) [4]. In computer vision, event cameras are often used in many fields, such as detection, tracking, recognition, and simultaneous localization and mapping (SLAM) [5,6].

In motion segmentation, independent motion-related pixels can be separated from the video sequence. Based on different motion types, the foreground objects are divided from the background by clustering these pixels [3,7]. Traditional vision-based processing algorithms usually assume that the video is shot under ideal circumstances [8,9,10]. Many factors are not taken into account by traditional vision tasks (e.g., moving speed, stability, and lighting conditions). In practical-application scenarios, the performance of motion-segmentation algorithms will be severely affected by fast-moving devices, such as drones and self-driving vehicles. The problem is well handled by motion segmentation based on event cameras [11]. The task of spatial-motion segmentation aims to separate the tracked feature points according to the respective rigid three-dimensional (3D) motion [12].

In this paper, we propose a framework by fusing the EDPA and 3D-motion-compensation approach to establish the spatial-motion-segmentation algorithm. The advantages of 3D information and motion segmentation are well combined by the proposed algorithm, which can not only satisfy the accuracy and stability requirements of vision works, but also separate the objects from the redundant background. The datasets of the experiments derive from the ESIM, which is an event-camera simulator [13]. After collecting the datasets, the pseudo-depth of the per-event is obtained by the EDPA in each frame of a pixel. The VWE is conducted by accumulating the warped events, and the iterative-clustering algorithm is presented to maximize the contrast in the VWE. The proposed algorithm has great segmentation effectiveness in low-speed- and high-speed-motion scenes, and in low-light and low-exposure scenes (e.g., high noon and night scenes). The main contributions of this paper are as follows:
(1)In order to inexpensively and conveniently extend 3D (2D plane (x,y) and time (t)) to 4D (3D space (x,y,z) and time (t)) for events without an RGB-depth-map (RGB-D) camera [14], or based on the light detection and ranging system (LiDAR), the EDPA is proposed for application in a variety of scenarios in this paper;(2)Compared with the traditional frame-based and plane-based methods [15,16,17], the proposed algorithm combines the EDPA and 3D-motion compensation to accomplish the spatial-motion segmentation. The proposed algorithm provides a more refined segmentation model, and the segmentation accuracy can be greatly improved;(3)The SSA-Gradient Ascent is presented to maximize the contrast in the VWE in this paper, which blends the advantages of SSA [18] and gradient ascent to obtain a better fitness value and faster convergence rate than other algorithms, such as PSO (see Section 4.2).

The remainder of this paper is organized as follows. In Section 2, we review some related works on depth estimation and motion segmentation. The mathematical model and main methods are presented in Section 3. Section 4 presents the process of the experiments. The conclusions are shown in Section 5.

## 2. Related Work

Depth estimation is one of the most important components of the EDPA. In [19], the authors introduced some of the high-accuracy and high-speed structured-light 3D-reconstruction methods. Deep learning enables us to accomplish visual tasks that are difficult to achieve by traditional geometry-based algorithms. Muhammad, K. et al. proposed a bidirectional long short-term memory (BiLSTM)-based attention mechanism with a dilated convolutional neural network (DCNN) to recognize the different human actions in videos [20]. A primary-prioritized recurrent deep reinforcement learning algorithm for dynamic-spectrum access based on cognitive-radio (CR) technology was proposed in [21]. In [22], a game-theory strategy was proposed to improve the energy-consumption performance of the MEC system. David et al. presented the first monocular depth estimation based on a convolutional neural network (CNN) [23]. Ibraheem et al. presented supervised learning without coupled geometric information [24]. To overcome the disadvantages of the RGB information-only-based method, an improved moving-object-detection method was proposed in [25]. Depth estimation based on event cameras has been gradually developed in recent years. In [26], the authors introduced the event-based multiview stereo (EMVS) to estimate semidense 3D structures with known trajectories. A novel 3D-reconstruction method was researched by Kim et al., which can perform real-time 3D reconstruction from a single hand-held event camera with no additional sensing [27]. In [28], a unifying framework was studied to solve several computer-vision problems with event cameras: motion, depth, and optical-flow estimation.

A variety of segmentation algorithms have been proposed by researchers to recognize and separate the objects from the background in the scene. Wang et al. proposed the EvDistill to learn a student network on the unlabeled and unpaired event data (target modality) via knowledge distillation (KD) [29]. In [30], the authors introduced a framework for the segmentation of human-motion sequences based on sensor networks. Li et al. presented a novel approach, which is the motion segmentation of 3D-active-contour registration [31]. Event cameras are also widely made use of in motion segmentation. Stoffregen et al. presented a motion-compensation-based approach to establish the image of warped events (IWE) for segmentation [10]. The dataset was proposed by Mitrokhin et al. to perform the motion segmentation of event cameras [32]. The authors developed a technique for generalized motion segmentation based on spatial statistics across timeframes [33]. In [34], Zhou et al. introduced an event-based motion-segmentation method with spatiotemporal graph cuts, which cast the problem as an energy minimization one involving the fitting of multiple motion models.

## 3. Main Methods

The objectives of our research are to extend 4D events and realize spatial-motion segmentation. We assume the prior knowledge of the scene that we cannot know. Because the information carried by the per-event is limited, the events in the set $\varepsilon ={\left\{e_{k}\right\}}_{k=1}^{N_{e}}$ are processed so that we can obtain enough information. The depth maps of the images are obtained by depth estimation. The initial motion parameter $\theta ={\left[\theta_{x},\theta_{y},\theta_{z}\right]}^{T}$ can be obtained by motion-flow estimation [35]. The contrast maximization is a framework that provides state-of-the-art results on several event-based computer-vision tasks [36]. A coherent motion is represented by a cluster. The problems we need to address are to classify the events into distinct clusters and maximize the contrast in the VWE.

### 3.1. Events from 3D to 4D by EDPA

The target of EDPA is to obtain an extra dimension (Z) for events. Firstly, the pre-pixel depths of $Frame_{1}$ and $Frame_{2}$ are derived by depth estimation as $Z_{frame_{1}}$ and $Z_{frame_{2}}$, respectively. As shown in Figure 1a, we calculate the depth of the per-event by Equation (1). At the same time, the depth (Z) and events coordinates (x,y) are not in the same coordinate system. The coordinates need to be normalized by transformation. Then, we transform the pixel coordinates and depth (Z) into the same coordinate system through a global transformation, as shown in Equations (2) and (3) and Figure 1b. Finally, we have an extra dimension, as shown in Equation (4):(1)$$Z_{k}=\frac{t_{k}-T_{frame_{1}}}{T_{frame_{2}}-T_{frame_{1}}}\left|Z_{frame_{2}}-Z_{frame_{1}}\right|$$
where $t_{k}$ is the time when the event (k) is triggered; $T_{frame_{j}}$ is the time of the frame (j) in a video; $Z_{frame_{j}}$ is the pixel depth on the frame (j), and $Z_{k}$ is the depth of the event (k);
(2)$$\frac{x_{k}}{X_{k}}=\frac{y_{k}}{Y_{k}}=\frac{f}{Z_{k}}$$
where $\left(x_{k},y_{k}\right)$ is the raw coordinate of the event (k); $\left(X_{k},Y_{k},Z_{k}\right)$ are the global coordinates of the event (k); f is the scale factor;

Writing Equation (2) in matrix form, we have:(3)$$Z_{k}\left[\begin{array}{c} x_{k}\\ y_{k}\\ 1 \end{array}\right]=\left[\begin{array}{ccc} f & 0 & 0\\ 0 & f & 0\\ 0 & 0 & 1 \end{array}\right]\left[\begin{array}{c} X_{k}\\ Y_{k}\\ Z_{k} \end{array}\right]$$
(4)$${\left\{e_{k}=\left(x_{k},y_{k},t_{k}\right)\right\}}_{k=1}^{N_{e}}\to {\left\{e_{k}=\left(X_{k},Y_{k},Z_{k},t_{k}\right)\right\}}_{k=1}^{N_{e}}$$
where $e_{k}$ is the event (k) in the set, and $N_{e}$ is the total number of events.

### 3.2. Motio-Flow Estimation

A method of voxel motion in space over time is described by the motion flow. The grayscale-invariance assumption is shown in Figure 2.
(5)$$I_{t_{1}}(x_{1},y_{1},z_{1},t_{1})=I_{t_{2}}(x_{2},y_{2},z_{2},t_{2})$$
where $I_{t_{k}}$ is the grayscale value at the time ($t_{k}$), and (x,y,z) are the coordinates of the voxel in space.

As shown in Figure 2, according to the grayscale-invariance assumption, the movement of a voxel from t to t+dt is shown in Equation (6):(6)I(x+dx,y+dy,z+dz,t+dt)=I(x,y,z,t)

When Taylor expansion is performed on the left side of the above equation and retains the first-order term, we will obtain:(7)$$I\left(x+dx,y+dy,z+dz,t+dt\right)\approx I\left(x,y,z,t\right)+\frac{\partial I}{\partial x}dx+\frac{\partial I}{\partial y}dy+\frac{\partial I}{\partial z}dz+\frac{\partial I}{\partial t}dt$$

Comparing the right sides of Equations (6) and (7), we have:(8)$$\frac{\partial I}{\partial x}dx+\frac{\partial I}{\partial y}dy+\frac{\partial I}{\partial z}dz+\frac{\partial I}{\partial t}dt=0$$

Both sides of the above equation are divided by dt, and $\frac{\partial I}{\partial t}$ is moved to the right side of the equation:(9)$$\frac{\partial I}{\partial x}\theta_{x}+\frac{\partial I}{\partial y}\theta_{y}+\frac{\partial I}{\partial z}\theta_{z}=-\frac{\partial I}{\partial t}$$
where $\theta_{x},\theta_{y},\theta_{z}$ are the velocities in the x,y,z directions, respectively.

To obtain enough equations to estimate the θ, three planes ($S_{1},S_{2},S_{3}$) are created as (x,y,t),(y,z,t),(z,x,t), respectively, and their grayscale values are $I_{1},I_{2},I_{3}$, respectively, which are shown in Figure 3. Equation (9) can be written as:(10)$$\frac{\partial I_{k}}{\partial i}\theta_{i}+\frac{\partial I_{k}}{\partial j}\theta_{j}=-\frac{\partial I_{k}}{\partial t}$$
where (i,j) is the element that belongs to {(x,y),(y,z),(z,x)}, and k is the index of the set that belongs to {1,2,3}. For example, when k = 1, (i,j)=(x,y), Equation (10) can be written as:(11)$$\frac{\partial I_{1}}{\partial x}\theta_{x}+\frac{\partial I_{1}}{\partial y}\theta_{y}=-\frac{\partial I_{1}}{\partial t}$$

Equation (10) can be written as a matrix form:(12)(I1,xI1,y00I2,yI2,zI3,x0I3,z)[θxθyθz]=−(∂I1∂t∂I2∂t∂I3∂t)
where $I_{1,x}$ is k = 1 and the partial derivative of $I_{1}$ to x.

$I_{k,\left(i,j\right)}$ are recorded as $a_{k}$ and $b_{k}$, and the above equation can be written as:(13)$$\left(\begin{array}{ccc} a_{1} & b_{1} & 0\\ 0 & a_{2} & b_{2}\\ b_{3} & 0 & a_{3} \end{array}\right)\left[\begin{array}{c} \theta_{x}\\ \theta_{y}\\ \theta_{z} \end{array}\right]=-\left(\begin{array}{c} c_{1}\\ c_{2}\\ c_{3} \end{array}\right)$$

By calculating the optical flow, $\theta_{x}$ can be obtained such that:(14)θ=(θx−a1θx+c1b1−b3θx+c3a3)=(θxθyθz)

### 3.3. 3D-Motion-Compensation and the Iterative-Clustering Algorithm

As shown in Figure 4, the objects are warped in the direction of the optical flow, and the image becomes sharp; in the contrary case, the image becomes a blur. The warped events are given by:(15)$$x_{k}={\left(x_{k},y_{k},z_{k}\right)}^{T}$$
(16)$$x_{k}^{\prime }=x_{k}-\left(t_{k}-t_{ref}\right)\theta =x_{k}-\Delta t_{k}\theta$$
where $t_{k}$ is the trigger time of the event (k), and $x_{k}^{\prime }$ is the position where the event ($x_{k}$) is warped.

Similar to the IWEs, we accumulate the warped events according to the weight values. The definition of the VWE is:(17)$$V_{j}\left(x\right)=\sum_{k=1}^{N_{e}}p_{kj}\delta \left(x-x_{kj}^{\prime }\right)$$
where $p_{kj}$ represents the probability that the event (k) belongs to the optical flow (j), and $\delta =\{\begin{array}{c} 1,x-x_{kj}^{\prime }=0\\ 0,otherwise \end{array}$ represents the Dirac function. $x_{kj}^{\prime }$ is the location, where the event (k) is warped along the optical flow (j).

Variance is employed to evaluate the contrast in this paper. To make the VWE sharper, the contrast is maximized as:(18)$$\mathrm{Var}\left(V_{j}\right)=\frac{1}{\left|\Omega \right|}{\int_{\Omega }\left(V_{j}\left(x\right)-\mu_{V_{j}}\right)}^{2}dx$$
where Ω is the image volume, and $\mu_{V_{j}}$ denotes the mean of the VWE over the image volume (Ω).

For the discrete VWE, the above equation can be written as:(19)$$\mathrm{Var}\left(V_{j}\right)={\sum_{a,b,c\in \Omega }\left(V_{abc}-\mu_{V_{j}}\right)}^{2}$$
where a,b,c is the index of the voxel in spacetime.

Now, the θ and P are needed to initialize and iterate. The elements of the P are the event-cluster membership probabilities, and the elements of the θ are the motion parameters. Because our clustering type is motion flow, the k-means algorithm was used to find the cluster center to initialize the θ. The loss function of the k-means algorithm is defined as follows:(20)$$J\left(c,\lambda \right)=\min\sum_{i=1}^{M}{\left\|x_{i}-\lambda_{c_{i}}\right\|}^{2}$$
where $x_{i}$ represents the sample point (i), c is the cluster to which $x_{i}$ belongs, $\lambda_{c_{i}}$ represents the center point corresponding to the cluster, and M is the total number of samples.

To make the per-event have the same probability of being classified into the cluster, we initialize the P:(21)$$P=\frac{1}{N_{l}}$$
where $N_{l}$ is the clusters number.

The above process can be summarized as the need to optimize the θ and P that maximize the variance, and the θ and P are updated by the iterative-clustering algorithm. Each iteration of the iterative-clustering algorithm has two steps: a fixed P to update the θ, and a fixed θ to update the P:(22)$$\left(\theta^{*},P^{*}\right)=\underset{\left(\theta ,P\right)}{\arg\max}\sum_{j=1}^{N_{l}}\mathrm{Var}\left(V_{j}\right)$$

The iteration of the θ makes use of the SSA-Gradient Ascent. In SSA, the producers’ location update is given by:(23)θi,jt+1={θi,jt⋅e(−iα⋅itermax)if R2<STθi,jt+q⋅lif R2≥ST
where t is the current iteration number, and θi,jt represents the value of the dimension (j) of the sparrow (i) at iteration (t). The largest number of iterations is represented by itermax.α∈(0,1], and q is a random number obeying a normal distribution. R2∈[0,1] represents the alarm value, and ST∈[0.5,1] represents the safety threshold (l∈ℝ1×d).

The scroungers are updated by:(24)θi,jt+1={q⋅e(θworstt−θi,jti2)if i>n2θpt+1+|θi,jt−θpt+1|⋅n+⋅lotherwise
where θp is the optimal position currently occupied by the producers; θworst represents the present global worst position; n∈ℝ1×d, and n+=nT(nnT)−1.

The sparrows that are aware of the danger are given by:(25)$$\theta_{i,j}^{t+1}=\{\begin{array}{cc} \theta_{best}^{t}+\beta \cdot \left|\theta_{i,j}^{t}-\theta_{best}^{t}\right| & if\;f_{i}>f_{g}\\ \theta_{i,j}^{t}+K\cdot \left(\frac{\left|\theta_{i,j}^{t}-\theta_{worst}^{t}\right|}{\left(f_{i}-f_{w}\right)+\varepsilon }\right) & if\;f_{i}=f_{g} \end{array}$$
where $\theta_{best}$ represents the present global best position; β is the step-size-controlled parameter; K∈[−1,1]; $f_{g}$ and $f_{W}$ are the current global fitness values, which represent the best and worst fitness values, respectively; ε is a constant that avoids zero in the denominator.

The SSA is used to find the better θ in the global area. Gradient ascent is used to obtain the best in the local area, as shown in (25).
(26)$$\theta \leftarrow \theta +\mu \nabla_{\theta }\left(\sum_{j=1}^{N_{l}}\mathrm{Var}\left(V_{j}\right)\right)$$
where μ is the iteration step size.

The iteration of the P is given by:(27)$$p_{kj}=\frac{c_{j}(x_{k}^{\prime }(\theta_{j}))}{\sum_{i=1}^{N_{l}}c_{i}(x_{k}^{\prime }(\theta_{i}))}$$
where $c_{i}$ is the VWE of the i-th; $x_{k}^{\prime }(\theta_{j})$ is the position where the event (k) is warped along the 3D-motion flow (j); $p_{kj}$ is the probability that the event (k) belongs to the j-th.

The algorithm flowchart is shown in Figure 5. The Algorithm 1 and Figure 5 show the overall process of our method. Variance convergence can usually be judged by the following conditions:(1)The variance curves tend to be horizontal, or the fluctuation changes a little;(2)In the variance values, more than a certain constant can be considered variance convergence;(3)The iterations will be interrupted when the number of iterations reaches a certain value.
**Algorithm 1.** Spatial-Motion Segmentation**Start****Input:** Raw events, set $\varepsilon_{1}={\left\{e_{k}=\left(x_{k},y_{k},t_{k},p_{k}\right)\right\}}_{k=1}^{N_{e}}$; original images; the number of clusters ($N_{l}$).**Output:** Event-cluster membership probabilities (P), and motion parameters (θ).**Procedure:****1:****For** each image, **do** depth estimation.
        **End for;**
**2: For**k←1 to $N_{e}$, **do:**           
$Z_{k}=\frac{t_{k}-T_{frame_{1}}}{T_{frame_{2}}-T_{frame_{1}}}\left|Z_{frame_{2}}-Z_{frame_{1}}\right|$

$Z_{k}\left[\begin{array}{c} x_{k}\\ y_{k}\\ 1 \end{array}\right]=\left[\begin{array}{ccc} f & 0 & 0\\ 0 & f & 0\\ 0 & 0 & 1 \end{array}\right]\left[\begin{array}{c} X_{k}\\ Y_{k}\\ Z_{k} \end{array}\right]$

       **End for:**

$\varepsilon_{2}={\left\{e_{k}=\left(X_{k},Y_{k},Z_{k},t_{k},p_{k}\right)\right\}}_{k=1}^{N_{e}}$
**3:****For**$e_{k}$, **do:**           computing $\theta_{k}$ according to $\theta ={\left(\theta_{x},\frac{-a_{1}\theta_{x}+c_{1}}{b_{1}},\frac{-b_{3}\theta_{x}+c_{3}}{a_{3}}\right)}^{T}$
       **End for:**
**4: For**k←1 to $N_{e}$, **do:**           warping $e_{k}$ according to $x_{k}^{\prime }=x_{k}-\left(t_{k}-t_{ref}\right)\theta =x_{k}-\Delta t_{k}\theta$
       **End for:**
**5: For** each warped $e_{k}$, **do** weighted accumulate:           
$V_{j}\left(x\right)=\sum_{k=1}^{N_{e}}p_{kj}\delta \left(x-x_{kj}^{\prime }\right)$
**6: For**i←1 to ∞, **do:**           **If** the variance is not converged, **then:**              update the P using $p_{kj}=\frac{c_{j}(x_{k}^{\prime }(\theta_{j}))}{\sum_{i=1}^{N_{l}}c_{i}(x_{k}^{\prime }(\theta_{i}))}$              update the θ according to SSA-Gradient Ascent:              computing $Var\left(V_{j}\right)={\sum_{a,b,c\in \Omega }\left(V_{abc}-\mu_{V_{j}}\right)}^{2}$           
**End if:**

i←i+1

       **End for:**
**End**


## 4. Experiments

The performance of the proposed algorithm was evaluated by two experiments. The following experiments were conducted to verify the feasibility of the proposed algorithm. We utilized our datasets, which were derived from the event-camera simulator (ESIM). Our experimental process consisted of two parts: in the first part, the spatial-motion segmentation was implemented in the exterior scene and interior scene, and in the second part, the influence of the optimization-algorithm performance on the variance in the VWEs was estimated by SSA-Gradient Ascent, PSO, and gradient ascent.

### 4.1. Experiment 1: Spatial-Motion Segmentation of Exterior and Interior Scenes

The Monodepth2 [38] depth-estimation network was utilized to predict the exterior scene in this paper, which was in a situation where the camera was stationary and only had moving objects. The predicted result is shown in Figure 6a, which displays the depth map of the first keyframe. The EDPA needs to utilize two images to expand the dimension of the events. After we obtained the depth map of the first key frame, the second key frame was selected for the depth estimation based on the trigger time of the 15,000th event. The spatial-motion segmentation was performed as shown in Figure 6b,c. Figure 6b is warped in the wrong direction, which is not the motion-flow direction of the objects. Figure 6c is the segmentation result by warping in the correct direction. When the algorithm iterates to around the 10th round, the variance has been maximized, and we can consider that the variance has converged. The iterative plots of variance are shown in Figure 6d. The changes in the θ are shown in Figure 6e–g. Six curves represent lateral-velocity vectors ($\theta_{x}$), column-velocity vectors ($\theta_{y}$) and velocity vectors on the z-axis ($\theta_{z}$).

The fully convolutional residual network (FCRN) [39] was applied to predict the interior scene, as shown in Figure 7a. In Figure 7b,c, two carts along the upper-left and lower-right corners of the table with different directions and speeds were successfully separated into different classes in 3D space, and little redundant background information was segmented into the foreground. Experimenting with the objects moving in different directions demonstrated the feasibility of the proposed algorithm in the interior scene.

### 4.2. Experiment 2: Comparison of the Effects of Different Optimization Algorithms

The SSA has a preferable ability in global optimization, while it is weak in local searches. The gradient ascent has an excellent performance in local optimization, but it easily falls into the local extreme value. The SSA-Gradient Ascent combines the advantages of SSA and gradient ascent. The updates of the θ by the SSA-Gradient Ascent, PSO, and gradient ascent were evaluated in this part, as shown in Figure 8. The variance-convergence rate of the algorithms and the value of the variance after convergence are listed in Table 1. As presented by the data in Table 1 and Figure 8, we chose the PSO method as the benchmark value. In Motion Flow 1, the proposed algorithm outperforms the basic value, with the gradient ascent 0.402% lower than the baseline. In Motion Flow 2, the gradient ascent is 0.731% higher than the basic value, while the proposed algorithm is 0.819% higher than the basic value. The same initial value of the θ is used by the above algorithms, and the maximum-variance value of the SSA-Gradient Ascent indicates that the proposed algorithm is less dependent on initialization. The gradient-ascent method does not converge until 19 iterative rounds. Compared with PSO, the proposed algorithm’s convergence rate was 52.941% higher in Motion Flow 1, and 46.154% higher in Motion Flow 2. The better performance of the proposed algorithm was confirmed according to this experiment.

## 5. Conclusions

A spatial-motion-segmentation algorithm is proposed in this paper, which has fused the EDPA and 3D-motion-compensation approach. The accuracy for tasks such as feature detection in complex environments is addressed by the proposed algorithm, while the advantages of 3D information and motion segmentation are well combined. The pseudo-depth of the per-event was obtained by the EDPA in each frame of a pixel. The VWE was conducted by accumulating the warped events, and a spatial-motion-segmentation algorithm is presented to maximize the contrast in the VWE. Interior and exterior scenes were segmented in the first part of the experiments. In the second part, the effect on the variance in the VWEs was estimated by SSA-Gradient Ascent, PSO, and gradient ascent. In Motion Flow 1, the SSA-Gradient Ascent was 0.402% higher than the basic variance value, and 52.941% faster than the basic convergence rate. In Motion Flow 2, the gradient ascent was 0.731% higher than the basic value, and 46.154% faster than the basic convergence rate. As a result, the experimental results validate the feasibility of the proposed algorithm. For future research, more complex scenes and conspicuous-spatial-motion segmentation will be studied by us.

## Figures and Tables

**Figure 1 sensors-22-06732-f001:**
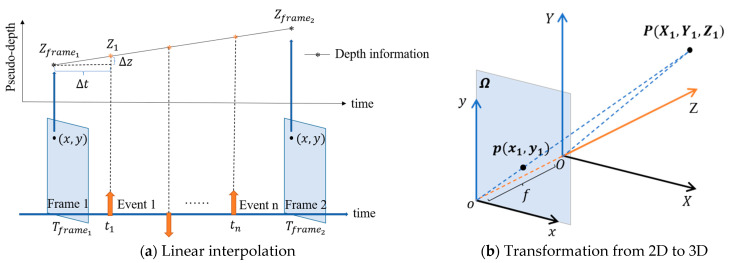
The events-dimensionality-preprocessing algorithm. (**a**) Two images are keyframes. The upward and downward arrows between the two images represent positive and negative events, respectively. (**b**) The point $p\left(x_{1},y_{1}\right)$ is on the Ω plane, and the p is back-projected into the global coordinate system O−X−Y−Z by coordinate transformation.

**Figure 2 sensors-22-06732-f002:**
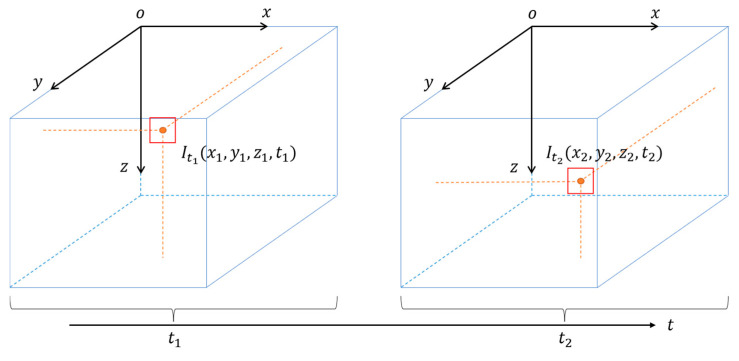
Grayscale-invariance assumption. According to the assumption, the grayscale value of the voxel is not changed between $t_{1}$ and $t_{2}$ in space.

**Figure 3 sensors-22-06732-f003:**
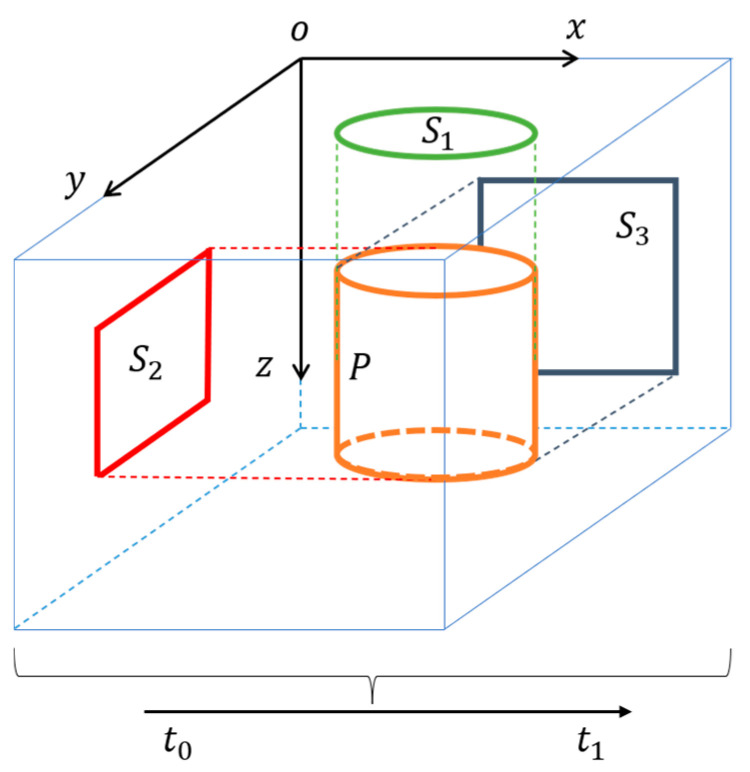
2D projection of a 3D time surface (TS) of events. The 3D TS is a 3D map in which each voxel stores a single time value [37]. The object (P) is obtained by accumulating events from $t_{0}$ to $t_{1}$, and then projecting it onto three planes ($S_{1},S_{2},S_{3}$ ) to form TSs in space.

**Figure 4 sensors-22-06732-f004:**
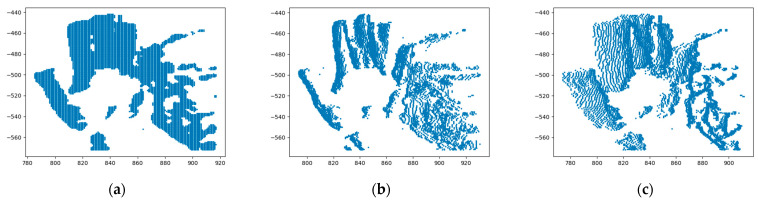
Images of warped events (IWEs): (**a**) TS when the events are not warped; (**b**) events are warped in the direction of the “hand”, and so the image of the “hand” becomes sharp, and the image of the “face” becomes a blur; (**c**) the condition presented by the image is counter to (**b**).

**Figure 5 sensors-22-06732-f005:**
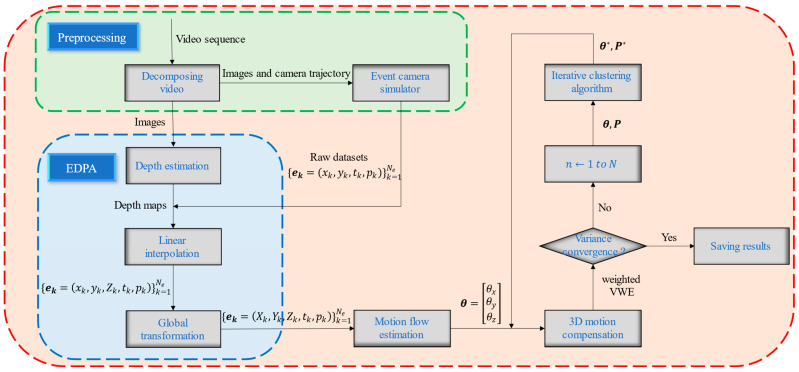
The spatial motion segmentation algorithm flowchart. The n is the current number of iterations. The N represents the total number of iterations required. In the whole algorithm, the dataset we need is obtained by the preprocessing part, and the extension of events from 3D to 4D is realized by the EDPA. The VWE is conducted by accumulating the warped events, and the iterative-clustering algorithm is used to maximize the contrast in the VWE.

**Figure 6 sensors-22-06732-f006:**
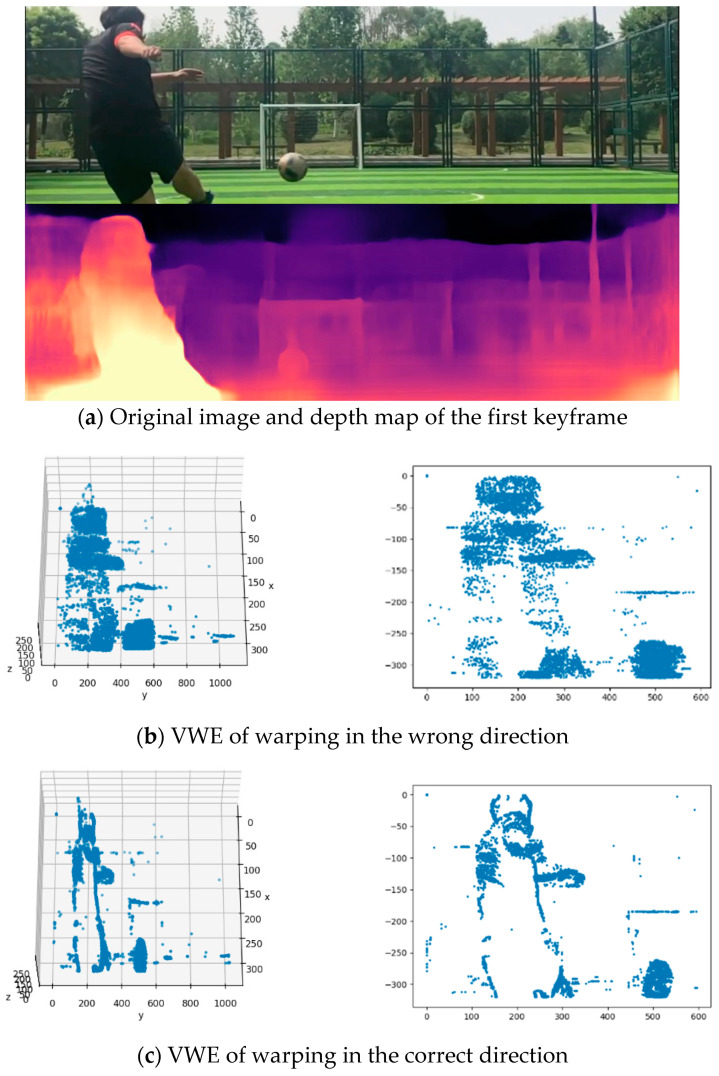
Results of exterior-scene experiment. (**a**) The left is an image of the original scene. The right is the predicted depth map. The time of the first frame comes from the first event’s trigger time. (**b**,**c**) Warping events to obtain the VWE. The initial value of the velocity vectors of warping events comes from the k-means algorithm. (**d**) Variance 1 and Variance 2 represent the variance change in the VWE after events are warped along with different motion flows. (**e**–**g**) Iterating process of motion parameters (θ).

**Figure 7 sensors-22-06732-f007:**
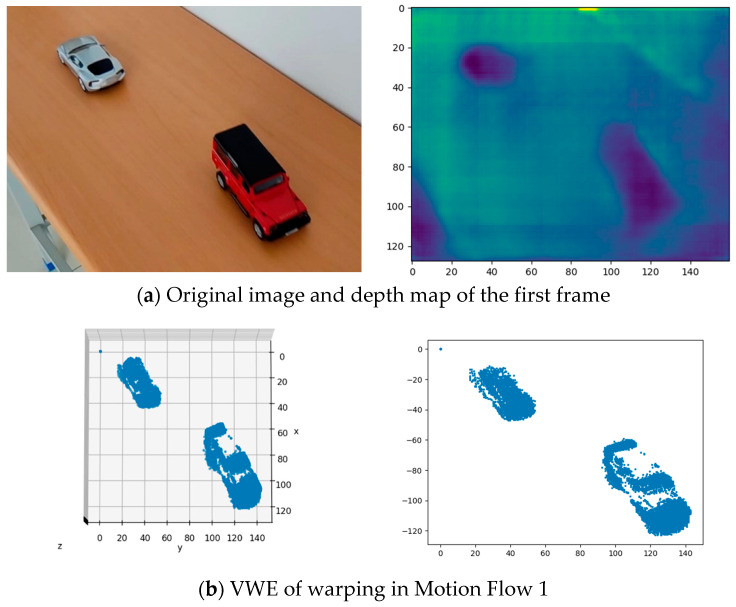
Results of interior-scene experiment. (**a**) Original image and predicted depth map. (**b**) Events are warped by the motion flow towards the lower-right corner. The red car is sharpened, and the silver car is blurred. (**c**) Events are warped by the motion flow towards the upper-left corner, the silver car being sharpened, and the red car being blurred. (**d**) Variance 1 and Variance 2 represent the variance change in the VWE after events are warped, along with different motion flows. (**e**–**g**) Iterating process of motion parameters (θ).

**Figure 8 sensors-22-06732-f008:**
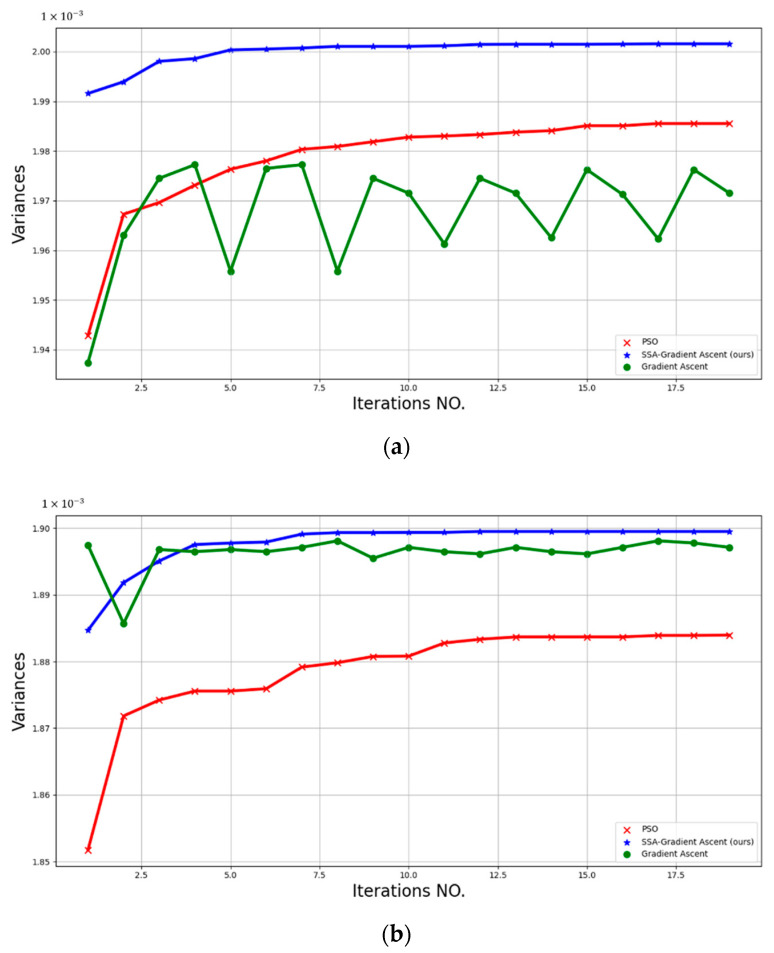
The variance convergence curves of VWEs when the θ is optimized by SSA-Gradient Ascent, PSO, and gradient ascent. (**a**) Variance curves along the direction of Motion Flow 1 (**b**) Variance curves along the direction of Motion Flow 2.

**Table 1 sensors-22-06732-t001:** Comparison of algorithm performances.

		**Motion Flow 1**		**Motion Flow 2**	
Variance value	**SSA-Gradient Ascent (ours)**	**0.00200104**	**+0.800%**	**0.00189911**	**+0.819%**
	PSO	0.00198522	0%	0.00188369	0%
	Gradient Ascent	0.00197723	−0.402%	0.00189746	+0.731%
Convergence speed (iteration no.)	**SSA-Gradient Ascent (ours)**	**8**	**+52.941%**	**7**	**+46.154%**
	PSO	17	0%	13	0%
	Gradient Ascent	None		None	

## Data Availability

Data sharing not applicable.

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
