# Peer review of "A Spatial-Motion-Segmentation Algorithm by Fusing EDPA and Motion Compensation"

_sensors, 2022, doi:10.3390/s22186732_

Round 1
Reviewer 1 Report
Major points:
1. The authors should add numerical improvements in the abstract against the SOTA.
2. contribution 2 and 3 have contradiction and repetition and the overall contribution is weak and needs more attention and improvement to show your novelty.
3. Sections 4.2 and 4.3 need more explanation, especially equations that have not had enough description and are difficult to understand.
4. Figure 5 is vague and not explained well.
5. Limitation of 2D and 3D is missing, it should be mentioned in the revised version with reference and justification.
6. In Sections 1 and 2, the authors must comment on the cited articles after introducing each relevant work. What readers require is, by convincing literature review, to understand the explicit thinking/consideration of why the proposed approach can reach more effective results. These are the very contribution of the authors.
7. Explain the reasons that the suggested approach provides better performance compared to other previous models? I wonder if the proposed method can be applied to other regions with different mechanical systems and occupant profiles.
8. The comparison table is missing. The authors must conduct some statistical tests to ensure the superiority of the proposed approach, i.e., how could the authors confirm that their results are superior to others? I strongly recommend adding a dedicated comparison table with other existing and popular systems.
9. The manuscript, however, does not link well with recent literature on recognition that appeared in relevant top-tier journals," Human action recognition using attention-based LSTM network with dilated CNN features” is missing it should be cited.
Section Conclusion - Authors are suggested to include in the conclusion section the real actual results for the best performance of their proposed methods in comparison towards other methods to highlight and justify the advantages of their proposed methods.
Author Response
Please see the attached file and thank you very much.

Reviewer 2 Report
This paper propose a spatial motion segmentation algorithm by fusing the Events Dimensionality Preprocessing Algorithm (EDPA) and the Volume of Warped Events (VWE). It is meaningful and interesting. However, some issues should be addressed or revised or improved below.
1.Figure 6 are made up of many figures, the original image in (a) and (b) is same. So it is repeated. Only one is ok. The figures should be encoded for each one for more clear reading.
2. The Figure 7, same problem as Figure 6.
3. Figure 8, why motion flow 1 and flow 2 are existed above the axis, it is not normal. The figure should be revised.
4. The method has been applied for Motion Segmentation, so could it also be applied for non Motion Segmentation and why?
5. What kinds of the application scenes that the method cloud be used, for example?
Author Response

(The authors gave the same response as above.)

Reviewer 3 Report
In this paper, the authors propose a spatial motion segmentation algorithm by fusing the Events Dimensionality Preprocessing Algorithm (EDPA) and the Volume of Warped Events (VWE). The EDPA consists of depth estimation, linear interpolation, and coordinates normalization to obtain an extra dimension Z of events. However, there are some major concerns as follows:
1. Abbreviations should be given full names when they first appear, such as ESIM, please check the full text carefully.
2. A Sparrow Search Algorithm-based Gradient Ascent is adopted to formulate the problem. Please further justify why this particular Gradient Ascent is suitable for the studied problem. Why did the author use gradient ascent instead of gradient descent?
3. In section 3 Problem Formulation, the authors did not provide a specific problem model, i.e., "problem formulation". It is suggested that the author should establish a mathematical model for the research problem and give a specific formula model to describe the research problem.
4. Clearly indicate the domain-independent innovative advance brought about by the proposed algorithm.
5. The current text is read for attuned experts. The article is already long, but a few tutorial paragraphs on the techniques that make up the algorithm would make following the text easier for many readers. At the authors' discretion.
6. This paper studies the optimization of image segmentation. There are many researches on Federated learning, such as "lightfed: an efficient and secure federated edge learning system on model splitting". Can you use the method of Federated learning?
7. The resolution of the experimental figures is too low to see the text clearly. It is suggested that the author provide the original figures with high resolution. The legends in figures 6 and 7 should use Greek letters instead of words.
8. There are also some typos in this paper. Please carefully go through the manuscript to improve its presentation.
Author Response
Please see the attched file and thank you very much.

Round 2
Reviewer 1 Report
Good Luck!
Author Response
We deeply appreciate the Reviewer for the time and e ort expended in the review of the paper, as well as for the encouraging positive comments of our work.
Reviewer 3 Report
The authors have addressed most of the concerns raised by the reviewers and their revisions have substantially improved the manuscript. However, there are still some minor issues to be addressed, namely:
1. Presentation is better but there is still room for improvement.
2. In Algorithm 1, it is recommended to use a positive sequence number.
3. In Algorithm 1, Output: VWEs. However, VWEs is not a variable and should not be the output value. In addition, there is no output process of VEWS in algorithm 1.
4. In the literature, there are some works on neural network. However, these refs are still too few. Authors are suggested to review more new and relevant research to support their research contribution. Some refs could be useful, e.g., RDRL: A Recurrent Deep Reinforcement Learning Scheme for Dynamic Spectrum Access in Reconfigurable Wireless Networks; A game-based deep reinforcement learning approach for energy-efficient computation in MEC systems.
5. The format of the paper shall be carefully checked.
6. Some references have incomplete compilation, e.g., missing volume/page numbering, such as ref [26].
Author Response
Please see the attached reply letter and thank you very much.
